# Research on multi-condition optimization of centrifugal compressor impeller meridian profile

Ning Yu[1], Xiaohan Yu[1]*, Zhi Cai[2], Xinle Yang[1], Liyong Tian[1], Song Li[3]

1 Mechanical Science and Engineering, Liaoning Technical University, Fuxin, Liaoning Province, China,
2 Huadian Liaoning Energy Development Co., Ltd. Shenyang Branch, Sujiatun, Liaoning Province, China,
3 School of Automobile and Traffic Engineering, Liaoning University of Technology, Jinzhou, Liaoning Province, China

* 1060592986@qq.com

## Abstract

The turbocharger, a pivotal technology for energy conservation and emission reduction, offers substantial academic significance, particularly in the in-depth study of its core component: the centrifugal compressor impeller. This research aims to enhance the centrifugal compressor's overall efficiency by optimizing its impeller meridional profile. By modifying the impeller meridional profile of a certain centrifugal compressor impeller to address the issue of discontinuous curvature, this paper aims to optimize the comprehensive average efficiency under variable speed and variable flow conditions. The optimization goal is to improve the overall average efficiency while maintaining the high pressure ratio based on the prototype scheme. The NSGA-III optimization algorithm is employed for multi-condition optimization, aiming to achieve comprehensive efficiency improvements under multiple operating speeds. The following conclusions are drawn: under multi-speed conditions, the optimized scheme exhibits comprehensive efficiency improvements over the prototype scheme, with an expanded stable operating flow range and maintained high-pressure ratio based on efficiency improvement, without sacrificing the operating flow range. Comparisons of internal flow conditions indicate that the optimized impeller features a smoother meridional passage and a reduced high-entropy region at the outlet, leading to lower entropy values. Additionally, pressure increases on both sides of the impeller blades while pressure differentials diminish, signifying enhanced internal flow conditions.

## 1 Introduction

The turbocharger, as an efficient and environmentally friendly product, plays a crucial role in contemporary road transportation, mobile machinery, and aerospace technology. The centrifugal compressor, as the core component of the turbocharger, holds significant importance in striving to achieve the "dual carbon" goals. Therefore, optimizing the design of centrifugal compressor impellers is of great significance.

In recent years, numerous scholars have conducted extensive research on the optimization design of centrifugal compressor impellers, primarily focusing on blade profile optimization

**Data availability statement:** The data to support this study can be found at the Figshare public repository - 10.6084/m9.figshare.279

**Funding:** National Natural Science Foundation of China [grant numbers 52174143].

**Competing interests:** The authors have declared that no competing interests exist.

and impeller passage optimization. Blade profile optimization design mainly concentrates on aspects such as blade curvature, angle, and sweep [1–4]. In addition to the impact of blade profile on the performance of centrifugal compressors, the impeller meridional profile also influences their performance. With the development of computer-aided design and manufacturing technology, there is a trend towards precision machining of impellers. Xu et al. [5,6] conducted research on impeller meridional profile of centrifugal impellers, indicating that optimizing the shape of impeller meridional profile can improve efficiency and pressure ratio without altering other components. Swain and Engeda [7,8] conducted FT and AT designs on impeller meridional profile. The results indicate that altering the meridional area of the passage can adjust the flow rate while maintaining the original pressure ratio, while changing the blade height at the impeller outlet can adjust the pressure ratio while maintaining the initial flow rate. Pakle et al. [9] employed Bessel curve for rim design and circular arc profile for hub design to improve the centrifugal compressor. The results indicate that through improved design, peak efficiency increased, and surge flow increased by 34%. Xinle Yang et al. [10] modified the hub profile of the impeller. The results indicate that as the curvature of the impeller hub increases, the impeller passage area increases, leading to an increase in the flow range and surge margin of the impeller. Additionally, the pressure ratio, efficiency, and temperature increase, while shock wave intensity and velocity loss decrease. Bonaiuti et al. [11] utilized Bessel curves for geometric parameterization and selected significant parameters through screening analysis to determine the most influential parameters. They then employed genetic algorithms for optimization. The results indicate that the efficiency of the impeller improved after optimization. Kim et al. [12] employed a genetic algorithm combined with a surrogate model to optimize four design variables of the impeller rim and hub. The results indicate that after optimization, both the isentropic efficiency and total pressure ratio improved. Ju Yaping et al. [13] utilized a combination of genetic algorithm (GA) and backpropagation (BP) to perform multi-objective optimization on 18 control points on the impeller rim and hub. The results indicate that after optimization, the internal flow conditions of the impeller improved, and the operating range was expanded. Li X et al. [14] proposed a multi-objective optimization strategy for centrifugal compressors combining data mining techniques with a time-series surge detection method. They applied this strategy to optimize the curvature of the sidewall and blades in the high-pressure ratio centrifugal impeller flow path. The results show that after optimization, both the total pressure ratio and the isentropic efficiency of the impeller significantly improved. Additionally, the surge margin and stall mass flow rate increased to varying degrees. Mojaddam and Pullen [15] utilized the response surface method to construct a surrogate model and identified optimal points based on the Box-Behnken method within the designed space. The results indicate that without sacrificing the operating range, optimizing the impeller efficiency and pressure ratio resulted in improvements compared to the prototype impeller. Ekradi et al. [16] integrated a three-dimensional blade parameterization method, genetic algorithm (GA), artificial neural network, and CFD solver to optimize the transonic centrifugal compressor impeller. The results demonstrate that under both design and off-design conditions, the optimized geometric performance surpassed that of the original impeller. Xinzi Tang et al. [17] proposed a multi-objective aerodynamic robust optimization and design exploration method for centrifugal compressor impellers. They constructed a neural network-based Kriging model and integrated it into aerodynamic robust optimization. The results show that after optimization, the average pressure ratio and average efficiency of the impeller were improved to varying degrees, while the sound power level decreased. Additionally, the neural network-based Kriging model exhibited higher accuracy in uncertain approximation modeling. Zhikai Chen et al. [18] utilized the Grey Relational Grade (GRG) multi-objective optimization method to find the optimal combination of one-dimensional parameters for the impeller. The results indicate that

the impeller outlet width and impeller outlet radius are the most sensitive parameters affecting compressor performance. The optimal impeller can reduce power consumption and improve isentropic efficiency at the design point. Jianqin Fu et al. [19] utilized the Reference Vector Guided Evolutionary Algorithm (RVEA) for multi-objective optimization of impeller structural parameters. The results indicate that after optimization, gas impact loss decreased, entropy increase decreased, and flow stability improved.

In current impeller meridional profile design schemes, other design parameters are typically considered, such as impeller outlet diameter and inlet/outlet flow angles. Additionally, some designs incorporate downstream diffuser sections, leading to conclusions that are not strictly attributed to changes in impeller meridional curvature. Analysis of optimization effects on centrifugal compressors is mainly confined to single-speed operating conditions, with insufficient investigation into internal flow dynamics under multi-speed conditions. As the impeller speed increases, the tip speed of the blades increases, resulting in a greater centrifugal force and compression effect. At low speeds, the airflow energy is insufficient, leading to lower pressure ratio and efficiency. However, at high speeds, although the pressure ratio increases, the efficiency decreases due to increased flow losses. Furthermore, optimization objectives are often restricted to the design speed, resulting in potential improvements at this speed but potential efficiency decreases in certain flow regimes under multi-speed conditions. Therefore, there is a need for multi-operating condition optimization of impeller meridional profile design parameters to achieve overall performance enhancement of centrifugal compressors.

It is worth noting that in some relevant studies, the simulation models often neglect the volute. However, the influence of the volute on the aerodynamic performance of centrifugal compressors cannot be overlooked, especially under off-design conditions where the impact of the volute on overall performance must be considered [20]. Therefore, it is necessary to conduct comprehensive simulation calculations of centrifugal compressors, which can lead to a more accurate design of optimal-performance centrifugal compressors.

## 2  Research object and numerical methods

### 2.1  Physical model

The three-dimensional model mainly consists of two parts: the impeller and the volute. Table 1 presents the main parameters of the prototype centrifugal compressor, while Fig 1 depicts the schematic diagram of the three-dimensional models of the impeller and the volute.

Table 1.  Main parameters table of prototype centrifugal compressor.

| Parameter | Numerical value |
| --- | --- |
| Design speed/(r/min) | 60000 |
| Number of main blade | 8 |
| Number of splitter blade | 8 |
| Diameter of impeller inlet hub/mm | 29.2 |
| Outer diameter of impeller inlet/mm | 92 |
| Diameter of impeller outlet/mm | 135.8 |
| Outlet width of impeller/mm | 8.67 |
| Outlet width of diffuser/mm | 6.77 |
| Diameter of volute outlet/mm | 69.9 |
| The inlet diameter of the diffuser/mm | 153.5 |
| The outlet angle of the impeller/° | 18.2 |
| Blade tip clearance/mm | 0.5 |

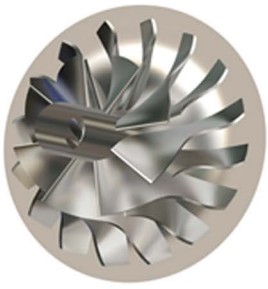
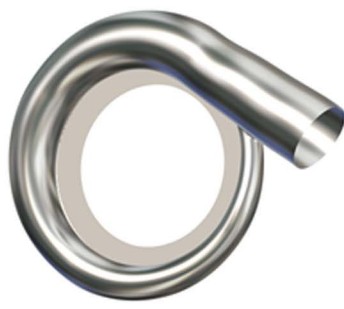

(a) Three dimensional model of impeller

(b) Three dimensional model of volute

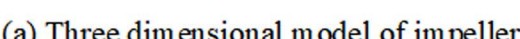

**Fig 1. Three-dimensional model diagram.**

## 2.2 Numerical methods

This paper employs the Euranus solver within the Openfine framework for numerical computations. Openfine facilitates concurrent computation and simulation of the internal gas flow within both the volute and impeller. The mesh utilizes non-matching face connections, enabling direct data transfer. The numerical solution is based on the Navier-Stokes equations, employing a central differencing scheme for spatial discretization within the computational domain. The solution is coupled with the Spalart-Allmaras (S-A) turbulence model for flow field resolution. Time advancement is accomplished using an explicit fourth-order Runge-Kutta method. To ensure computational accuracy and mitigate numerical errors, both second-order and fourth-order Runge-Kutta methods are employed. Additionally, CPU-Booster technology is utilized to accelerate computation speed and achieve convergence in computational results.

Based on the actual working environment of the centrifugal compressor, the internal fluid medium is set as ideal air with a temperature of 293 K. Simultaneously, the reference temperature and reference pressure are respectively set to 293 K and 101325 Pa. To conduct comprehensive numerical simulation analysis, the rotational speed is set from 40000 RPM to 600000 RPM, with increments of 10000 RPM. In the model setup, a completely non-matching connection is used for the dynamic stationary interface between the volute and impeller to ensure simulation accuracy and reliability. Utilizing the characteristics of the centrifugal compressor as a rotating axial intake mechanical device, a cylindrical coordinate system is adopted, with the intake direction set as the positive Z-axis. The pressure of the gas medium is set to 101325 Pa. Considering that the gas flow velocity inside the centrifugal compressor generally does not exceed the speed of sound, subsonic conditions are selected. The inlet boundary condition is set as Vz extrapolated. The turbulence model chosen is the Spalart-Allmaras model, with turbulent viscosity set to 0.0001 $\mathrm{m}^2$/s.

## 2.3 Mesh independence and experimental validation

The centrifugal compressor test was conducted on a fully automatic turbocharger performance test bench. This test bench utilizes an external air source to supply compressed air and simulates engine operating conditions through combustion chamber heating to drive the turbine, thereby rotating the impeller of the centrifugal compressor for performance testing. The test bench mainly consists of six parts: lubrication oil system, controllable air source, combustion chamber gas heating system, measurement and data acquisition system, control

system, and test bench piping system. The connection between the turbocharger test bench and the turbocharger test system pipeline is shown in Fig 2.

By utilizing the impeller periodic boundary conditions, the number of blades is set during the computational setup and a single-pass computational method is chosen, which can significantly reduce the amount of computation and the required computational resources. At the same time, it is guaranteed that other factors such as the model grid structure and numerical calculation settings are identical. Seven different mesh sizes are selected to compare the numerical simulation results. As shown in Fig 3, when the number of grids increases to 2.28 million, the numerical simulation results of isentropic efficiency and total pressure ratio remain constant. To ensure high computational accuracy while preventing excessive computational costs, the number of grids is set to 2.28 million, with 1.25 million grids for the impeller and 1.03 million grids for the volute.

The centrifugal compressor characteristic curves obtained from the above-mentioned experiments were compared with the characteristic curves obtained from the original numerical simulations of the compressor. The comparison of the efficiency and pressure ratio characteristics at multiple speeds is shown in Fig 3. Overall, the simulated results of the centrifugal

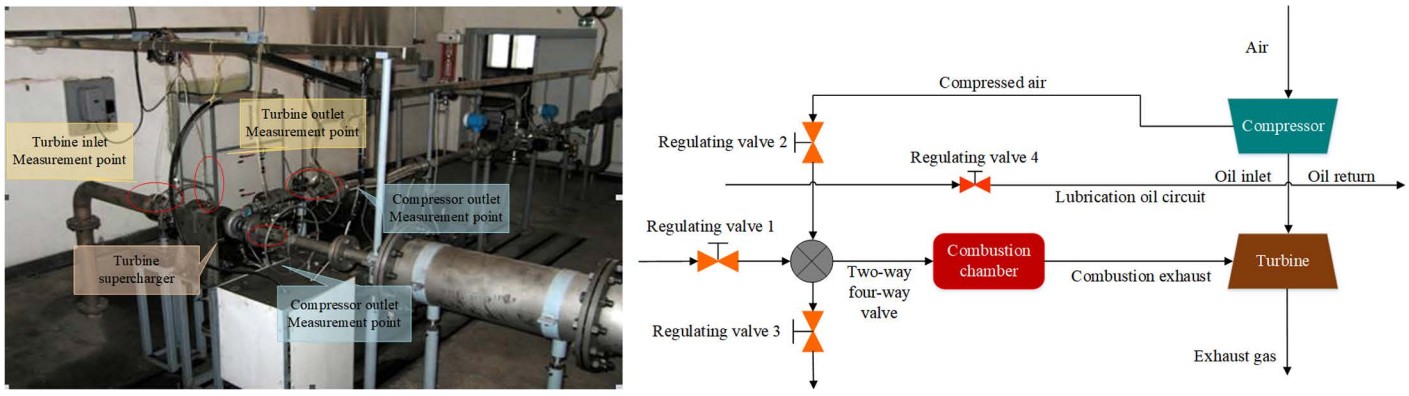

**Fig 2. Turbocharger test bench diagram & Turbocharger test system pipeline connection diagram.**

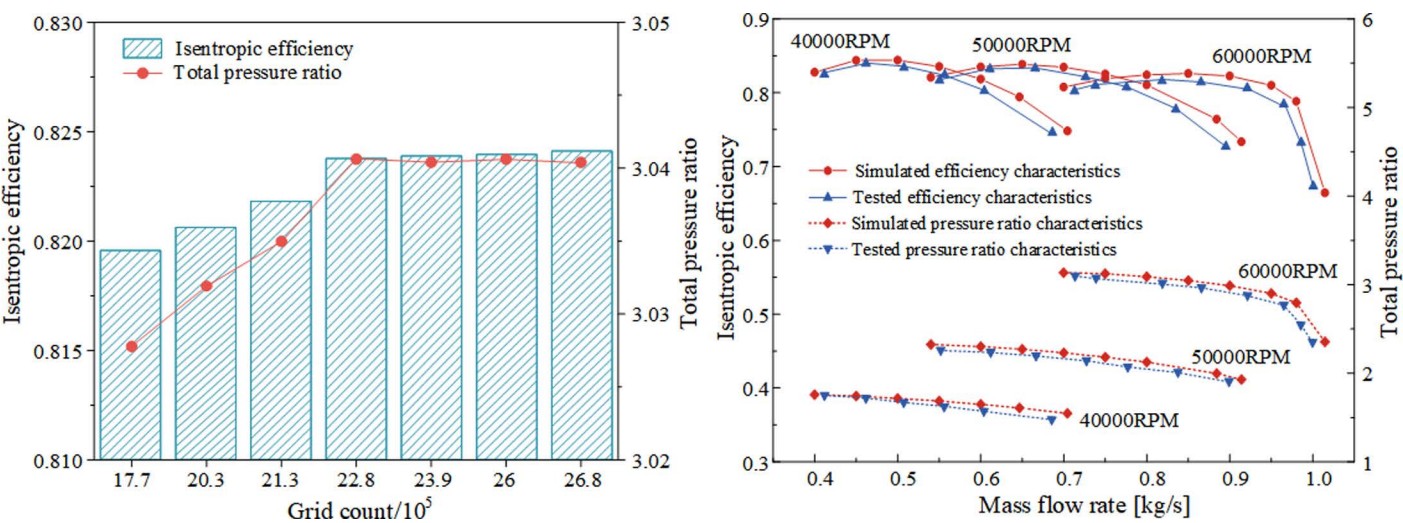

**Fig 3. Grid independence verification diagram and comparison diagram of simulation data and experimental data results.**

compressor were higher than the experimental results at different speeds (40000 RPM, 50000 RPM, and 60000 RPM), with the maximum error between the numerical simulation efficiency and pressure ratio and the experimental results being less than 5%. This is mainly because the fact that the boundary conditions for numerical calculations are all ideal conditions, neglecting factors such as heat transfer and surface roughness of the impeller. However, at the three operating speeds, the numerical simulation calculations effectively predicted the trends in the centrifugal compressor characteristics. Considering the various sources of errors in numerical simulations and the overall agreement in trends, it is concluded that the experimental data can verify that the numerical calculation methods used in this paper can predict the performance characteristics of the centrifugal compressor reasonably well [21].

# 3 Optimization of impeller meridional profile under multiple operating conditions

## 3.1 Design scheme and optimization targets

The optimized design scheme is shown in Fig 4. where $I_1$ and $O_1$ represent the inlet and outlet endpoints of the blade profile, respectively. Points $P_1$ and $P_2$ are controlled to move along the segments $I_1X_1$ and $O_1X_1$ originating from the inlet and outlet endpoints, respectively, by adjusting parameters s1/S1 and s2/S2. The movement of point $P_3$ along $I_1X_1$ is controlled to

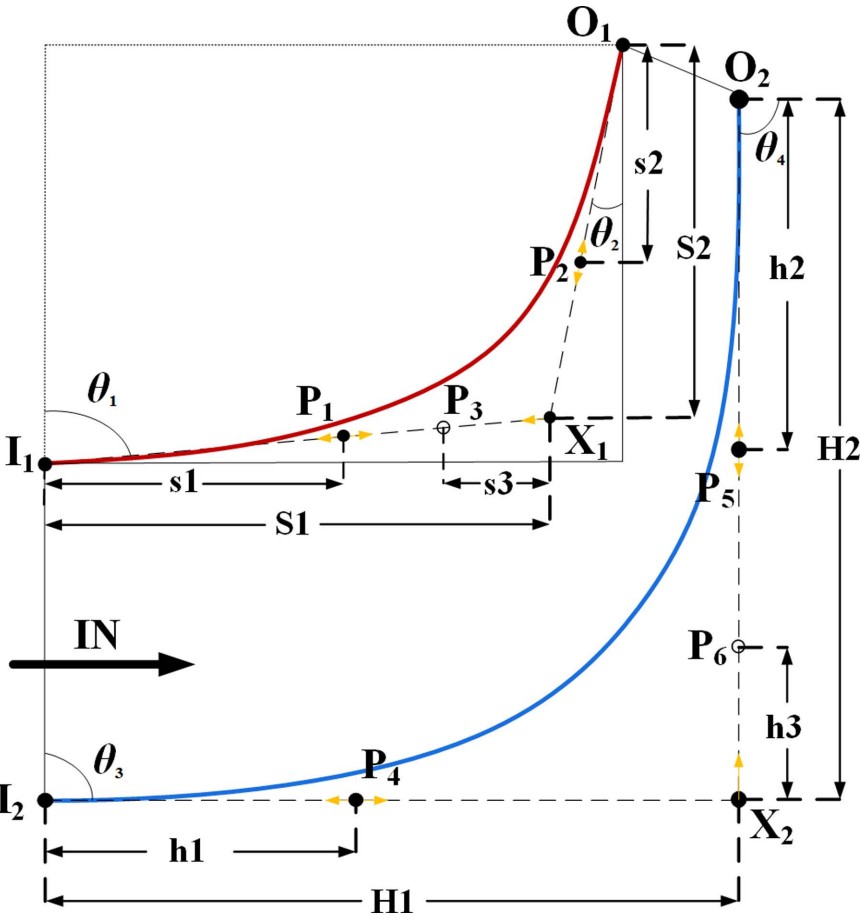

**Fig 4. The schematic diagram of key geometric parameters for impeller meridian line structure optimization.**

change the parameter s3/S1, thereby altering the blade profile. Similarly, $I_2$ and $O_2$ represent the inlet and outlet endpoints of the hub profile, respectively. Points $P_4$ and $P_5$ are controlled to move along the segments $I_2X_2$ and $O_2X_2$ originating from the inlet and outlet endpoints, respectively, by adjusting parameters h1/H1 and h2/H2. The movement of point $P_6$ along $O_2X_2$ is controlled to change the parameter h3/H2, thereby altering the hub profile. The angles $\theta_1$, $\theta_2$, $\theta_3$, and $\theta_4$ maintain consistency with the inlet and outlet flow angles of the impeller meridional channel and are set to 90°, 13°, 0°, and 0°, respectively. The design solution is for the shroud profile and hub profile only, and the overall extension and curvature of the blade remains the same as the prototype solution.

The optimization objectives and constraints are shown in Formulas (1) and (2) as follows:

$$\eta_{ave} \begin{cases} \eta_{ave6} = 0.5\eta_{0.90} + 0.5\eta_{0.75} \\ \eta_{ave5} = 0.5\eta_{0.80} + 0.5\eta_{0.65} \\ \eta_{ave4} = 0.5\eta_{0.60} + 0.5\eta_{0.45} \end{cases} \tag{1}$$

$$\pi_{ave} \begin{cases} \pi_{ave6} = 0.5\pi_{0.90} + 0.5\pi_{0.75} \\ \pi_{ave5} = 0.5\pi_{0.80} + 0.5\pi_{0.65} \\ \pi_{ave4} = 0.5\pi_{0.60} + 0.5\pi_{0.45} \end{cases} \tag{2}$$

In the formula, $\eta_{ave6}$, $\eta_{ave5}$, and $\eta_{ave4}$ are the comprehensive average efficiency at 60000RPM, 50000RPM, and 40000RPM speed respectively, $\eta_{ave} = 0.5\eta_{large\ flow} + 0.5\eta_{small\ flow}$, $\eta_{large\ flow}$, $\eta_{small\ flow}$ are the isentropic efficiency of large flow condition and the isentropic efficiency of small flow condition at each speed respectively; $\pi_{ave6}$, $\pi_{ave5}$, and $\pi_{ave4}$ are the comprehensive average pressure ratios at 60000 RPM, 50000 RPM, and 40000 RPM speeds, respectively. $\pi_{ave} = 0.5\pi_{large\ flow} + 0.5\pi_{small\ flow}$, $\pi_{large\ flow}$, $\pi_{small\ flow}$ are the total pressure ratio of large flow conditions and the total pressure ratio of small flow conditions at each speed.

As shown in Fig 5, single-factor analysis is conducted for each optimization variable, which means that when analyzing the impact of a certain optimization variable on the optimization objective, the remaining variables are set to their respective median values. Based on the above analysis results, the influence of each variable on the extreme values of $\eta_{ave6}$, $\eta_{ave5}$, and $\eta_{ave4}$ within the given range is analyzed. Within the range, the corresponding maximum change rates of $\eta_{ave6}$, $\eta_{ave5}$, and $\eta_{ave4}$ are calculated respectively. The calculation results are shown in Fig 5, where $S_1$ represents the sensitivity of each variable to the extreme values of $\eta_{ave6}$, $S_2$ represents the sensitivity of each variable to the extreme values of $\eta_{ave5}$, and $S_3$ represents the sensitivity of each variable to the extreme values of $\eta_{ave4}$.

According to the results, it is evident that the blade control parameter s1/S1 has a relatively high impact on $\eta_{ave6}$, while s2/S2 has the greatest impact on $\eta_{ave5}$ and $\eta_{ave4}$, followed closely by s3/S1. However, its impact on $\eta_{ave6}$ at the design speed is minimal. s3/S1 has a relatively balanced impact on all three optimization objectives. Among the hub control parameters, h1/H1 and h2/H2 have a greater impact on $\eta_{ave6}$ at the design speed compared to h3/H2.

Given that $\eta_{ave6}$ at the design speed is the primary optimization objective, and based on the above analysis, s1/S1, s3/S1, h1/H1, and h2/H2 are selected as the optimization variables. The remaining design parameters are set to their optimal values at the median. Additionally, the optimal range for the optimization variables is selected based on Fig 5, as shown in Table 2.

## 3.2 Optimization sample library

The Latin hypercube sampling method is selected to establish the optimized sample library. For the multi-condition optimization problem, the fitting and response are more accurate. Based on the literature [22], the number of basic tests required by the optimized Latin

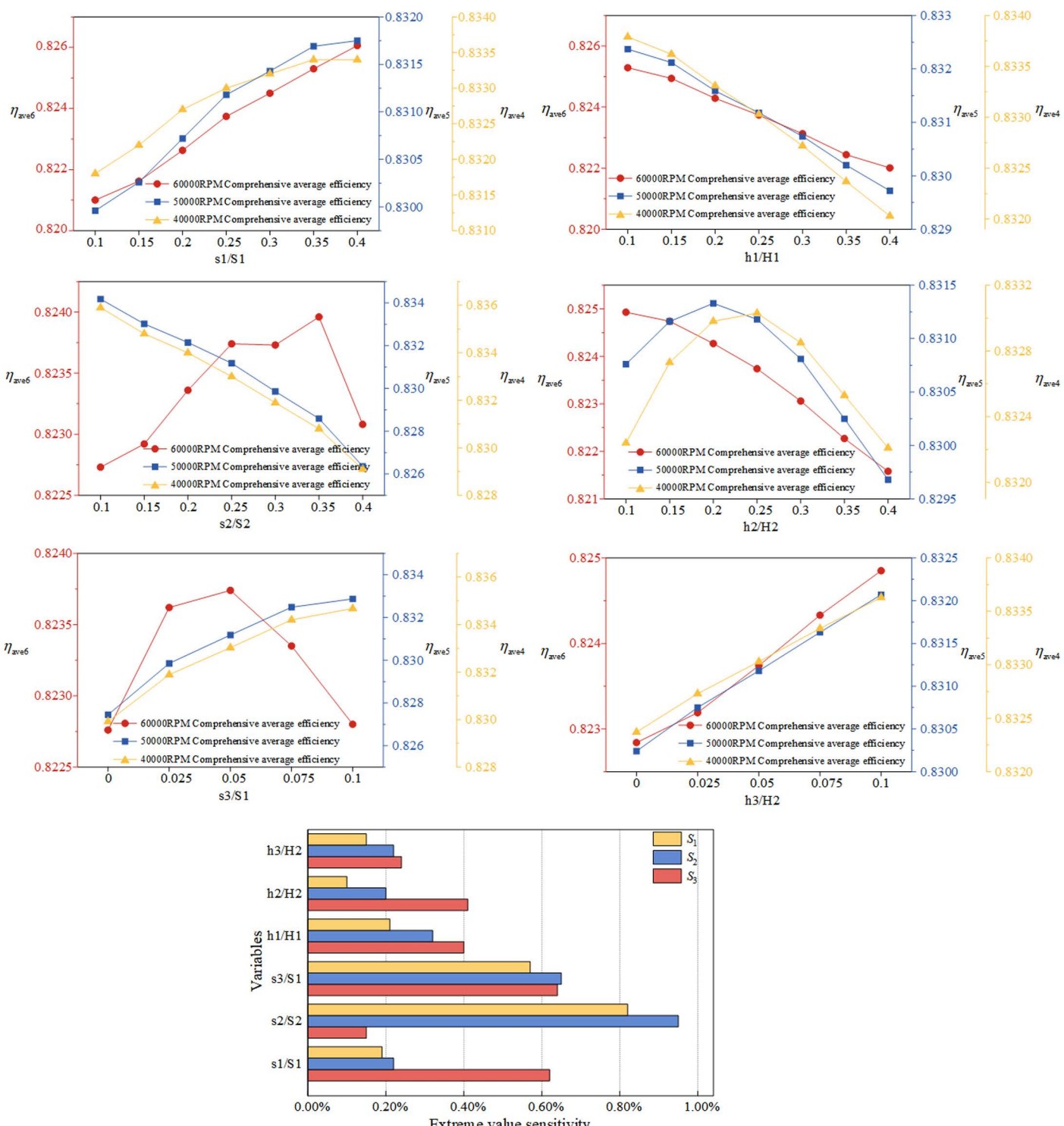

**Fig 5. The influence diagram of single factor change of optimization variables on optimization objectives & Variable extreme sensitivity distribution map.**

hypercubic sampling method is $((n+1)(n+2))/2$, where n indicates the number of irrelevant variables to each other, and the number of impeller meridian profile control variables is 4, the number of tests should be selected to be 15, and in order to make the results more accurate, 2 times the number of tests are selected, in addition, in order to validate the accuracy of the subsequent fitted regression model, another 5 tests are added as a verification samples, a total of 35 groups of tests were designed. The corresponding performance parameters obtained by numerical simulation are shown in Table 3.

### 3.3 Constructing multivariate nonlinear regression prediction model

The multivariate nonlinear regression prediction model is represented by equation (3):

$$y = b_0 + b_1 x_1 + \cdots + b_k x_k \tag{3}$$

In the formula, $y$ represents the optimization objectives and constraints. $b_k$ denotes the estimated value of the $k$ regression coefficient, while $x_k$ represents the $k$ term, where each term can be a single predictor variable, polynomial, or interaction term. In this paper, $x_1$, $x_2$, $x_3$, and $x_4$ correspond to s1/S1, s3/S1, h1/H1, and h2/H2, respectively.

$$\eta_{ave6} = 0.81181 + 0.0855 x_1 - 0.0027 x_2 - 0.00028 x_3 - 0.008875 x_4 - 0.0943 x_1^2 \\ - 0.1293 x_2^2 - 0.048 x_1 \times x_3 + 0.1541 x_2 \times x_3 \tag{4}$$

$$\eta_{ave5} = 0.82304 + 0.00733 x_1 + 0.0994 x_2 - 0.01299 x_3 + 0.04225 x_4 - 0.02069 x_1^2 \\ - 0.5344 x_2^2 + 0.0294 x_3^2 - 0.0831 x_4^2 + 0.1683 x_1 \times x_2 - 0.0584 x_2 \times x_3 \\ - 0.0896 x_2 \times x_4 \tag{5}$$

**Table 2. Optimize the variable name and level range table.**

| Variables | Range |
|---|---|
| s1/S1 | 0.25 ~ 0.40 |
| s3/S1 | 0 ~ 0.10 |
| h1/H1 | 0.10 ~ 0.25 |
| h2/H2 | 0.15 ~ 0.30 |

**Table 3. Optimized Latin hypercube test scheme table.**

| | s1/S1 | s3/S1 | h1/H1 | h2/H2 | $\eta_{ave6}$ | $\eta_{ave5}$ | $\eta_{ave4}$ | $\pi_{ave6}$ | $\pi_{ave5}$ | $\pi_{ave4}$ |
|---|---|---|---|---|---|---|---|---|---|---|
| 1 | 0.332 | 0.049 | 0.171 | 0.213 | 82.60% | 83.20% | 83.33% | 3.0828 | 2.2093 | 1.7004 |
| 2 | 0.380 | 0.074 | 0.119 | 0.261 | 82.66% | 83.40% | 83.51% | 3.0902 | 2.2134 | 1.7021 |
| 3 | 0.309 | 0.035 | 0.226 | 0.174 | 82.52% | 83.04% | 83.20% | 3.0767 | 2.2063 | 1.6991 |
| 4 | 0.358 | 0.065 | 0.142 | 0.229 | 82.65% | 83.33% | 83.44% | 3.0867 | 2.2113 | 1.7011 |
| 5 | 0.275 | 0.009 | 0.184 | 0.297 | 82.32% | 82.83% | 83.06% | 3.0781 | 2.2086 | 1.7009 |
| ...... | | | | | | | | | | |
| 25 | 0.366 | 0.056 | 0.168 | 0.235 | 82.63% | 83.26% | 83.39% | 3.0855 | 2.2108 | 1.7011 |
| 26 | 0.280 | 0.016 | 0.196 | 0.243 | 82.41% | 82.91% | 83.11% | 3.0775 | 2.2080 | 1.7004 |
| 27 | 0.400 | 0.073 | 0.126 | 0.170 | 82.76% | 83.39% | 83.46% | 3.0802 | 2.2077 | 1.6991 |
| 28 | 0.338 | 0.042 | 0.153 | 0.274 | 82.55% | 83.15% | 83.30% | 3.0848 | 2.2109 | 1.7014 |
| 29 | 0.351 | 0.067 | 0.130 | 0.243 | 82.64% | 83.35% | 83.46% | 3.0886 | 2.2124 | 1.7016 |
| 30 | 0.305 | 0.027 | 0.187 | 0.212 | 82.51% | 83.01% | 83.18% | 3.0770 | 2.2071 | 1.6998 |

$$\begin{aligned}
\eta_{\text{ave4}} = {}& 0.823568 + 0.0072x_1 + 0.08031x_2 + 0.00001x_3 + 0.04281x_4 \\
& - 0.02591x_1^2 - 0.3362x_2^2 + 0.01272x_3^2 - 0.08104x_4^2 + 0.13877x_1 \times x_2 \\
& + 0.02116x_1 \times x_4 - 0.0353x_2 \times x_3 - 0.1167x_2 \times x_4 - 0.02619x_3 \times x_4
\end{aligned} \tag{6}$$

$$\begin{aligned}
\pi_{\text{ave6}} = {}& 3.03073 - 0.06329x_1 + 0.4363x_2 + 0.071x_3 + 0.289x_4 \\
& - 1.917x_2^2 - 0.438x_4^2 + 0.542x_1 \times x_2 - 0.78x_2 \times x_3
\end{aligned} \tag{7}$$

$$\begin{aligned}
\pi_{\text{ave5}} = {}& 2.18608 - 0.01965x_1 + 0.0766x_2 - 0.0352x_3 + 0.1745x_4 \\
& + 0.1499x_3^2 - 0.2687x_4^2 + 0.1469x_1 \times x_2
\end{aligned} \tag{8}$$

$$\begin{aligned}
\pi_{\text{ave4}} = {}& 1.68173 + 0.00485x_1 + 0.02427x_2 + 0.02415x_3 + 0.09891x_4 \\
& - 0.0157x_4^2 + 0.0371x_1 \times x_2 - 0.0494x_1 \times x_3
\end{aligned} \tag{9}$$

Equations (4) to (9) represent the fitted regression models. The regression equation $R^2$ and $R^2$ adj were all above 95%, indicating a good fit. As illustrated in the Fig 6, the fitted values closely match the calculated values, demonstrating a nearly linear relationship with high agreement.

Through equations (4) to (9), the verification scheme in Table 4 was subjected to predictive calculations to validate the regression prediction model. From the Table 4, it is evident that the established regression prediction model exhibits small prediction errors for $\eta_{\text{ave6}}$, $\eta_{\text{ave5}}$, $\pi_{\text{ave6}}$, $\pi_{\text{ave5}}$, and $\pi_{\text{ave4}}$, all within 0.3%. The prediction error of $\eta_{\text{ave4}}$ is relatively large, but it is still within 1.2%, which is in line with the acceptable values commonly used in the engineering design field [23]. Thus, equations (4) to (9) can be considered as models for optimization objectives and constraints.

## 3.4  Optimization objectives and constraints

$$Max \begin{cases} \eta_{\text{ave6}}(x_1, x_2, x_3, x_4) \\ \eta_{\text{ave5}}(x_1, x_2, x_3, x_4) \\ \eta_{ave4}(x_1, x_2, x_3, x_4) \end{cases} \tag{10}$$

$$S.t. \begin{cases} \eta_{\text{ave6}} > 0.8214 \\ \eta_{\text{ave5}} > 0.8251 \\ \eta_{\text{ave4}} > 0.8318 \\ \pi_{\text{ave6}} > 3.056 \\ \pi_{\text{ave5}} > 2.198 \\ \pi_{\text{ave4}} > 1.695 \\ 0.25 < x_1 < 0.40 \\ 0.00 < x_2 < 0.10 \\ 0.10 < x_3 < 0.25 \\ 0.15 < x_4 < 0.30 \end{cases} \tag{11}$$

The multi-operating condition optimization objectives and constraints are represented by equations (10) and (11). Firstly, we define the optimization objective function. In this paper, achieving a comprehensive improvement in isentropic efficiency under multi-speed operating conditions compared to the original model is the ultimate goal. Therefore, in the parameter optimization process, we consider maximizing the values of the comprehensive average

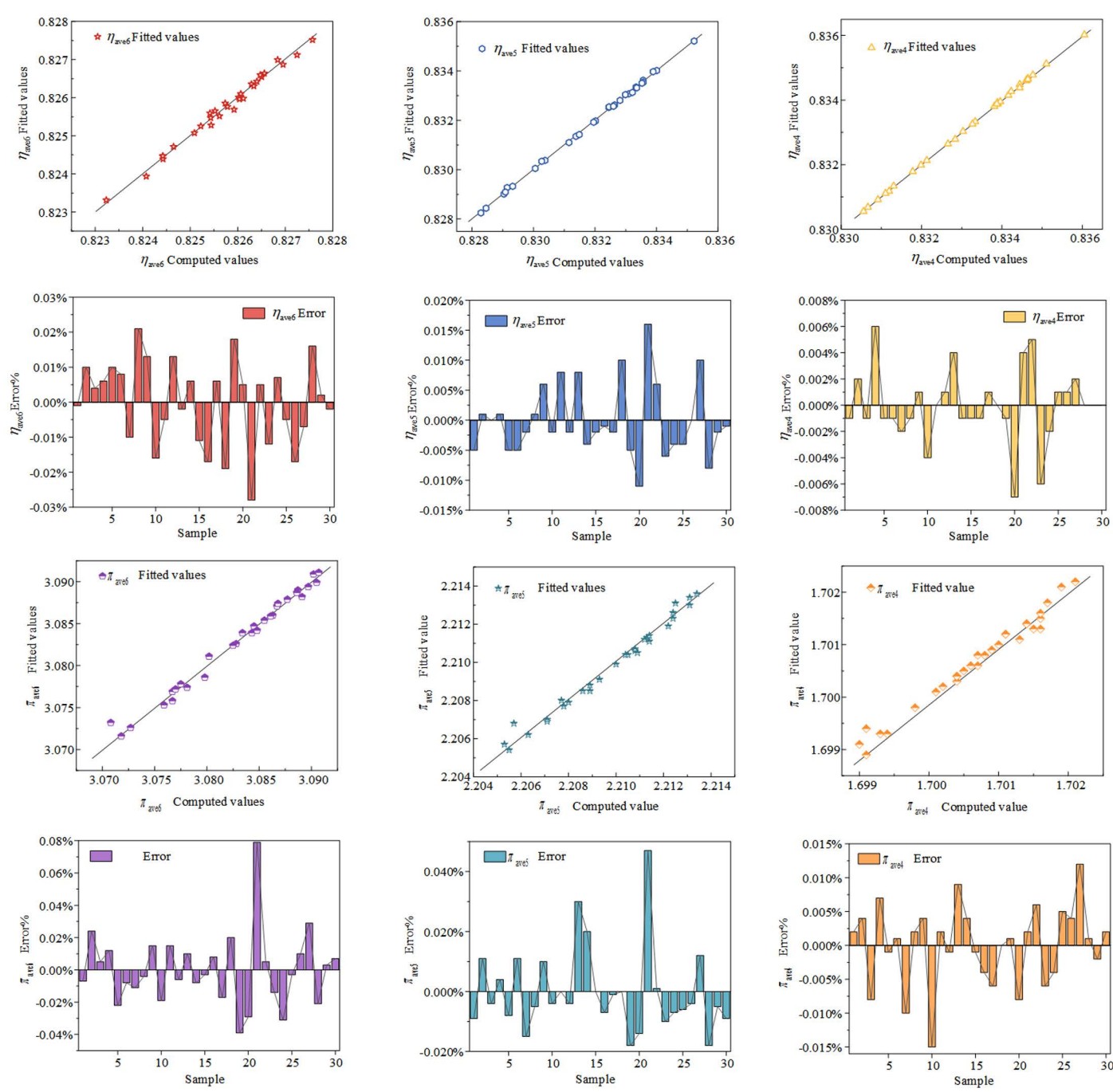

**Fig 6. The relationship between fitted values and calculated values, along with the error plot.**

efficiencies, $\eta_{ave6}$ at 60000 RPM, $\eta_{ave5}$ at 50000 RPM, and $\eta_{ave4}$ at 40000 RPM, as the optimization objectives. The regression prediction models $\eta_{ave6}$, $\eta_{ave5}$, and $\eta_{ave4}$ are used as the objective functions. Secondly, we create the constraint conditions to ensure that the optimization results meet the desired targets and achieve the required performance under relatively low speeds while maintaining a high-pressure ratio. Hence, the comprehensive average efficiencies $\eta_{ave6}$, $\eta_{ave5}$, and $\eta_{ave4}$, as well as the comprehensive average pressure ratios $\pi_{ave6}$, $\pi_{ave5}$, and $\pi_{ave4}$, from

**Table 4. Verification scheme table.**

| | $x_1$ | $x_2$ | $x_3$ | $x_4$ | $\eta_{ave6}$-error | $\eta_{ave5}$-error | $\eta_{ave4}$-error | $\pi_{ave6}$-error | $\pi_{ave5}$-error | $\pi_{ave4}$-error |
|---|---|---|---|---|---|---|---|---|---|---|
| 1 | 0.332 | 0.049 | 0.171 | 0.213 | 0.09% | 0.11% | 0.98% | 0.19% | 0.13% | 0.04% |
| 2 | 0.380 | 0.074 | 0.119 | 0.261 | 0.07% | 0.11% | 1.07% | 0.20% | 0.19% | 0.15% |
| 3 | 0.309 | 0.035 | 0.226 | 0.174 | 0.02% | 0.13% | 0.94% | 0.29% | 0.23% | 0.14% |
| 4 | 0.358 | 0.065 | 0.142 | 0.229 | 0.12% | 0.03% | 0.93% | 0.05% | 0.04% | 0.00% |
| 5 | 0.305 | 0.027 | 0.187 | 0.212 | 0.07% | 0.02% | 0.79% | 0.02% | 0.02% | 0.03% |

the original model at 60000 RPM, 50000 RPM, and 40000 RPM respectively, are considered as constraint conditions. Lastly, we establish the parameter optimization model. The objective of optimizing the impeller meridional profile parameters is to design the impeller with reasonable values for the four control variables $x_1$, $x_2$, $x_3$, and $x_4$. This ensures that the comprehensive average efficiencies $\eta_{ave6}$, $\eta_{ave5}$, and $\eta_{ave4}$ of the centrifugal compressor reach their maximum values at 60000 RPM, 50000 RPM, and 40000 RPM, respectively.

## 4  Optimization results and analysis

The third generation of Non-dominated Sorting Genetic Algorithm (NSGA-III) is a more advanced algorithm further proposed on the basis of NSGA-II. As shown in Fig 7, by introducing the reference point mechanism and using fast non-dominated sorting and efficient selection strategies, NSGA-III exhibits better convergence and diversity in solving optimization problems with three and more objectives compared to NSGA-II [24,25]. Currently, the NSGA-III algorithm has been widely applied in the engineering domain to tackle various nonlinear optimization problems [26–28]. Therefore, in this study, NSGA-III is adopted to solve the optimization problem of multi-operating conditions for the impeller involute profile parameters. The Pareto optimal solution set obtained through NSGA-III optimization and the impeller meridional profile are depicted in Fig 8. By comparing the optimized solution sets, the performance of this optimization can be summarized as follows: the values of $\eta_{ave5}$ and $\eta_{ave4}$ are relatively higher while $\eta_{ave6}$ is comparatively lower, and the optimization range of $\eta_{ave6}$ is relatively smaller. Therefore, prioritizing $\eta_{ave5}$ and $\eta_{ave4}$ as the primary selection objectives while considering $\eta_{ave6}$, an optimized solution that compromises higher values is chosen as the final optimization scheme.

### 4.1  Characteristics curve comparative analysis

Figure 9 presents a comparison of the characteristic curves of the compressor before and after optimization. From Fig 9(a), it can be observed that the efficiency of the optimization results is generally improved within the stable operating range compared to the prototype at 40000 RPM, 50000 RPM, and 60000 RPM operating conditions. The maximum efficiency increases by 0.86%, 1.00%, and 0.42%, respectively. Regarding the stable operating flow range, there are increases of 3.93%, 2.46%, and 2.84% at 40000 RPM, 50000 RPM, and 60000 RPM operating conditions compared to the prototype. From Fig 9(b), it is evident that there is little difference in pressure ratio at 40000 RPM and 50000 RPM operating conditions, while at 60000 RPM operating conditions, the optimization results show a slight increase compared to the prototype. From the above analysis, it can be concluded that the optimization of the impeller involute profile widens the stable operating range of the compressor. The efficiency under multiple speed conditions is superior to the prototype within the stable operating range, and there is also a slight increase in pressure ratio. Therefore, the optimization results achieve the expected objectives.

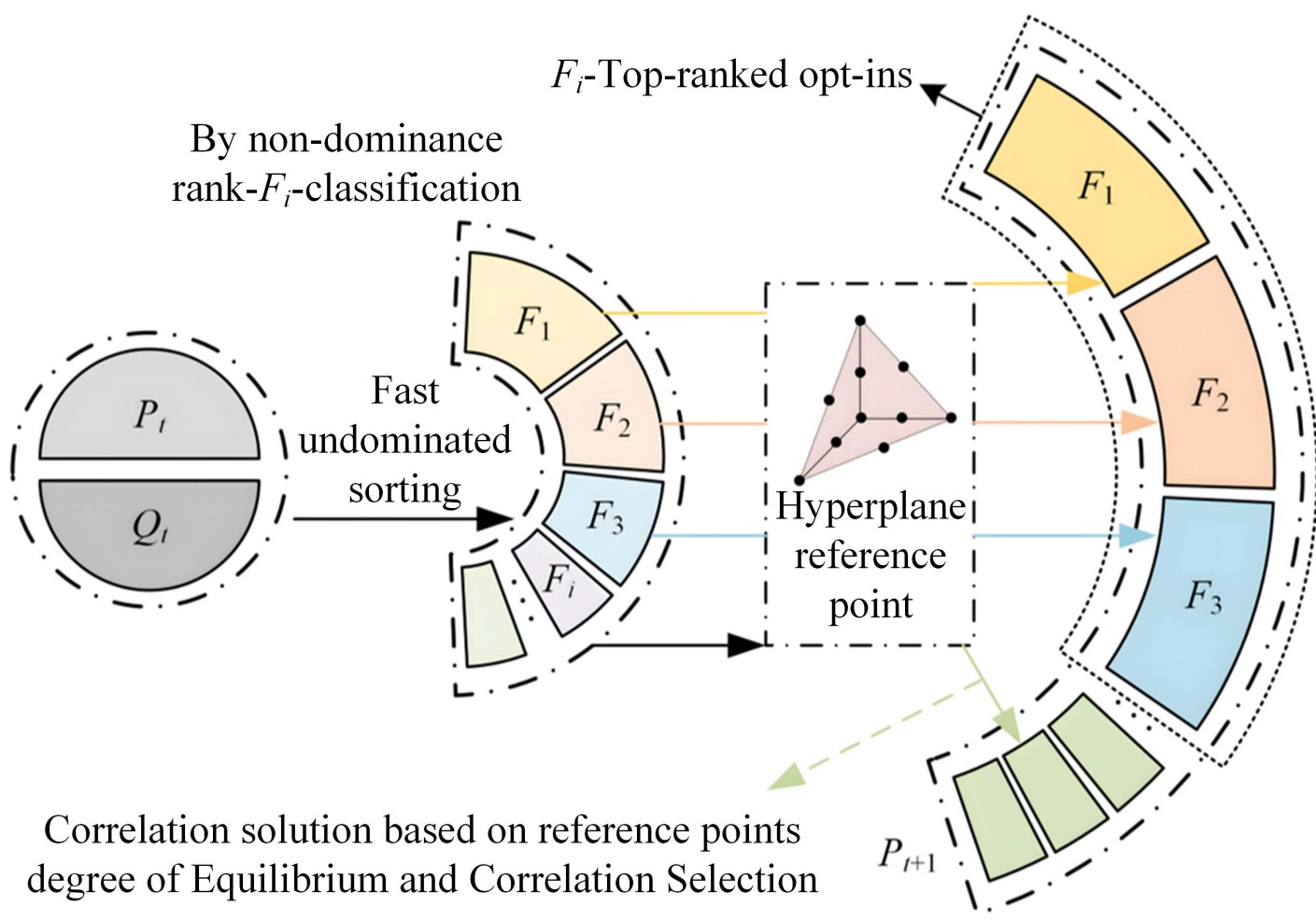

**Fig 7. NSGA-III basic framework diagram.**

## 4.2 Internal flow comparative analysis

From the analysis above, it can be inferred that due to the curvature of the impeller passage, centrifugal force, and adverse pressure gradient, the high-entropy region within the impeller mainly concentrates near the shroud tip. As the flow progresses through the passage, entropy values peak at the impeller exit. With an increase in rotational speed, the flow rate within the passage increases, while being influenced by leakage at the shroud tip and friction losses at the blade tip. Consequently, flow losses within the passage also increase. According to Fig 10, a comparison is made between the pre- and post-optimized impeller meridional passage entropy distributions and entropy extremums at the highest efficiency operating conditions. Post-optimization, the curvature of the impeller shroud transition from axial to radial is continuous and smooth, which helps to suppress secondary flows near the blade tip and leakage between blade tips. Under three different rotational speed conditions, the high-entropy region near the impeller shroud tip is significantly reduced after optimization. The entropy extremums within the impeller meridional passage are reduced by 14.37%, 14.17%, and 13.62% at rotational speeds of 40000 RPM, 50000 RPM, and 60000 RPM, respectively, thereby greatly reducing flow losses within the passage and improving the overall efficiency of the centrifugal compressor.

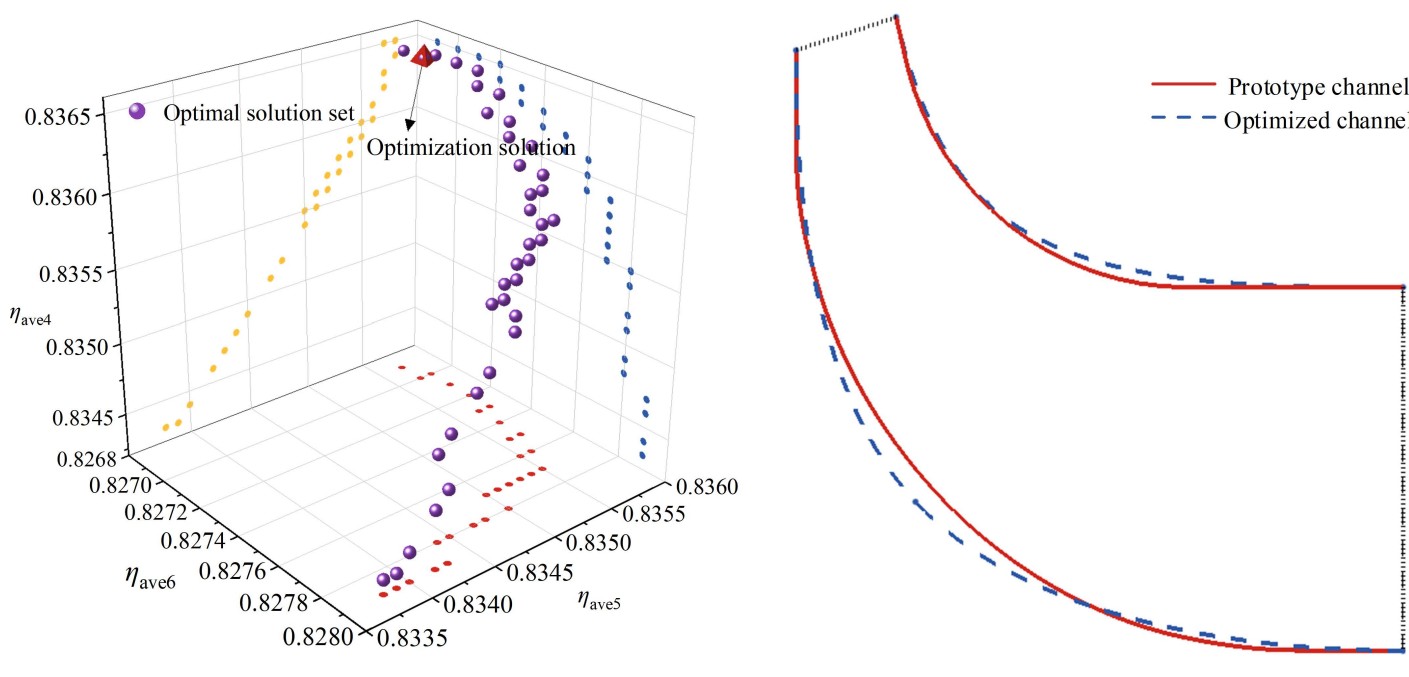

**Fig 8. Pareto optimal solution set diagram.**

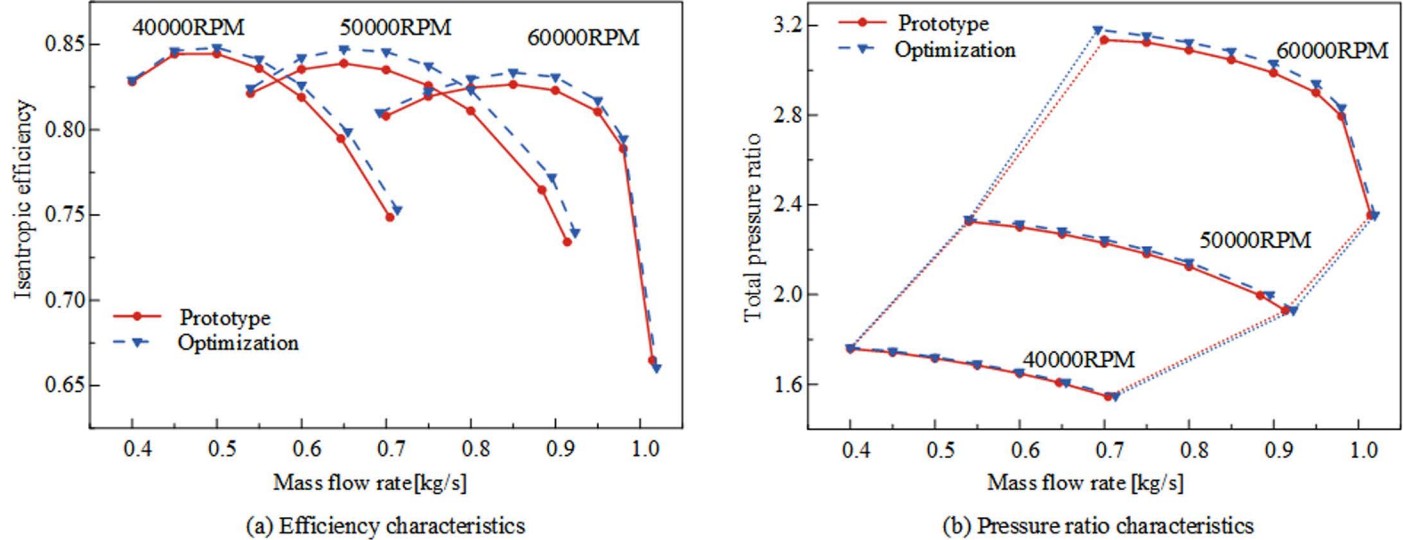

**Fig 9. Comparison of multi-speed characteristic curves of centrifugal compressor before and after optimization.**

According to Fig 11, a comparison is made between the pre- and post-optimized impeller outlet entropy distributions and entropy extremums at the highest efficiency operating conditions. Under three different rotational speed conditions, the original design exhibits high-loss regions concentrated at the suction side blade tip of the main blades and splitter blades, with relatively smaller losses on the pressure side, resulting in significant flow non-uniformity at the impeller outlet. Post-optimization, the flow non-uniformity at the impeller outlet is effectively suppressed under all three rotational speed conditions. The

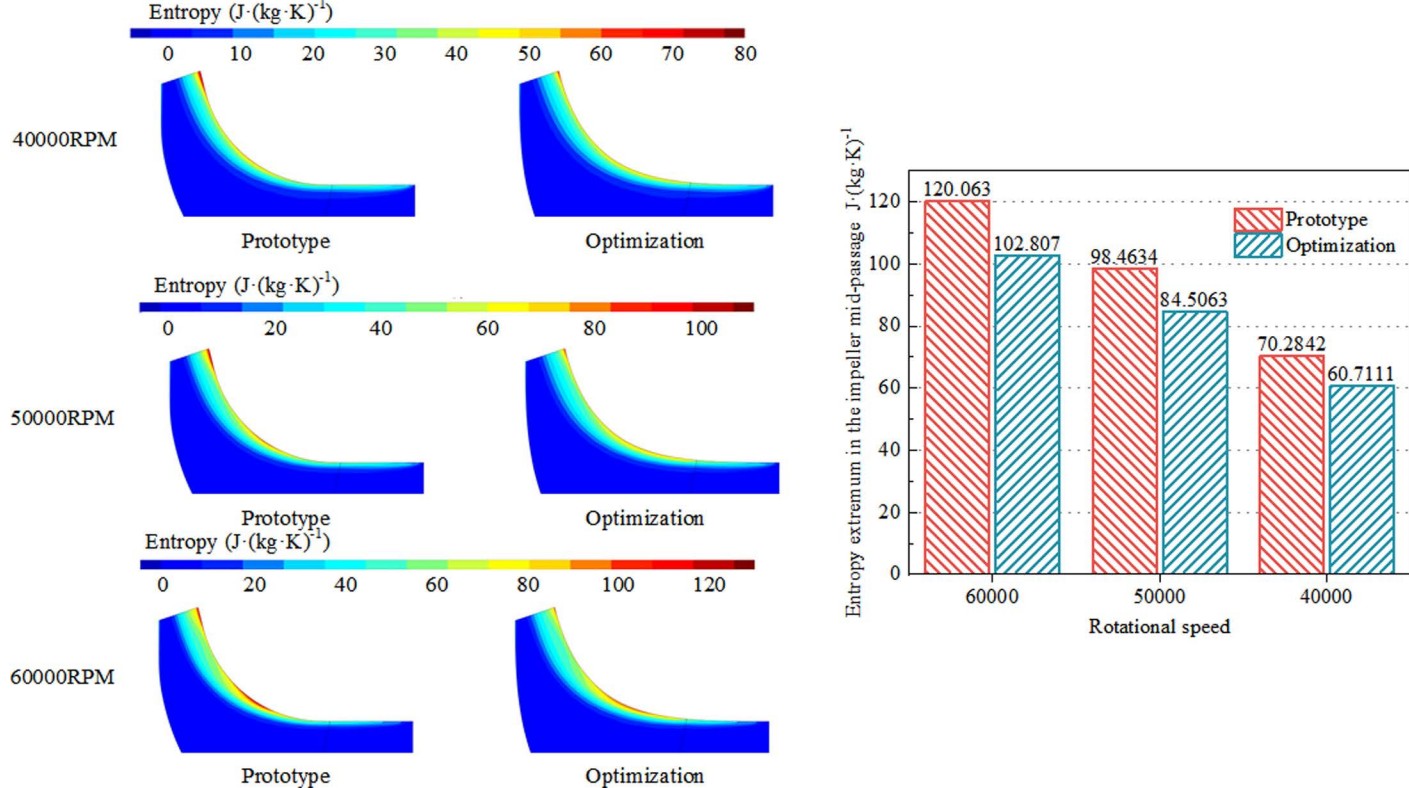

**Fig 10. The entropy distribution and entropy extreme value diagram of impeller meridian channel under the highest efficiency condition before and after optimization.**

entropy values on the suction side are noticeably reduced, leading to a decrease in the mixing losses between low-energy fluid and the mainstream. Under the three rotational speed conditions, the entropy extremums at the impeller meridional outlet are reduced by 6.35%, 4.60%, and 3.30% at rotational speeds of 40000 RPM, 50000 RPM, and 60000 RPM, respectively, resulting in a substantial reduction in impeller outlet flow losses. Overall, the optimized impeller design achieves significant improvements in enhancing the uniformity of the impeller outlet flow field and reducing flow losses, thereby enhancing the overall performance of the centrifugal compressor.

According to Fig 12, compare the static pressure distribution at 90% blade height of the main blades before and after optimization. Under three different rotational speeds, the static pressure distribution at the leading edge of the blades tends to be consistent before and after optimization. After optimization, the work area of the impeller passage increases, and there is a significant increase in gas pressure rise in the downstream section of the flow passage. Additionally, the static pressure values on the pressure and suction sides of the blades increase after optimization. Under three different rotational speeds, the prototype impeller's meridional curvature is discontinuous, and it is also affected by leakage losses in the tip clearance region, resulting in significant static pressure fluctuations. As the rotational speed increases, the airflow velocity increases, leading to an increase in pressure fluctuation, which is unfavorable for stable pressure rise in the impeller. In contrast, the meridional curvature of the impeller after optimization is continuous, weakening the acceleration phenomenon at the leading edge of the blade and reducing the pressure difference

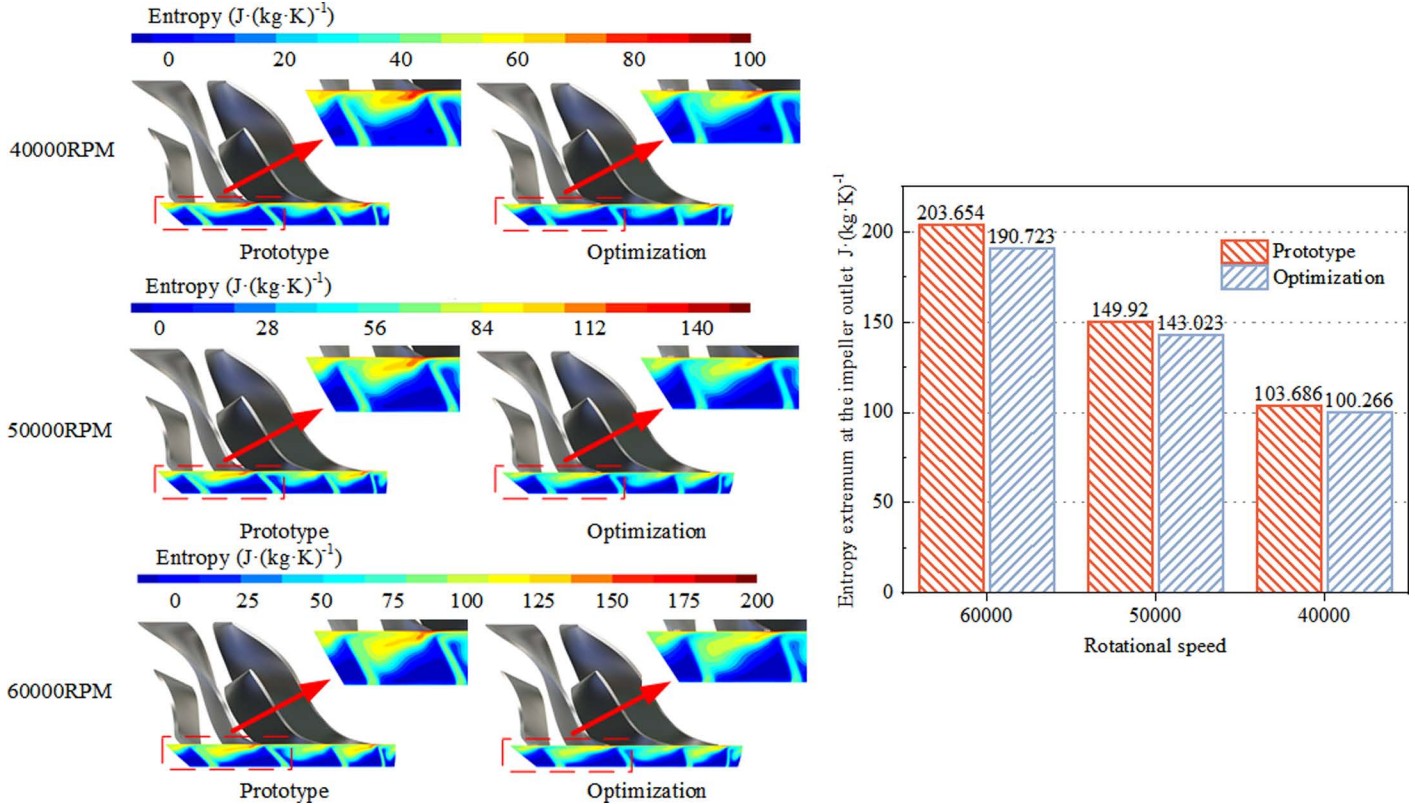

**Fig 11. The entropy distribution and entropy extreme value diagram of impeller outlet under the highest efficiency condition before and after optimization.**

on both sides of the blade, thereby weakening the transverse pressure gradient in the flow passage. This helps to prevent lateral flow in the passage and reduce leakage losses in the tip clearance. Furthermore, after optimization, the static pressure rise from the leading edge to the trailing edge becomes smoother, with reduced fluctuation, thereby enhancing the stability of the centrifugal compressor operation.

According to Fig 13, compare the distribution of relative Mach numbers at 90% blade height before and after optimization. The boxes in the figure represent the range of low-energy flow clusters where the relative Mach number is below 0.3. It can be observed from the figure that before and after optimization, shock waves are present at the leading edge of the suction side of the blades, and the low Mach number region is mainly concentrated in the downstream region of the impeller passage. Due to the curvature of the impeller flow passage and the centrifugal force, the fluid inside the passage is affected, leading to the aggregation of low-energy fluid downstream, forming a large-scale low-energy flow cluster. From 40000RPM to 60000RPM, the intensity of shock waves increases, and the range of the low-energy flow cluster with Mach numbers below 0.3 decreases. Compared to the prototype design, the meridional curvature of the impeller after optimization is continuous, suppressing flow separation and leakage losses at the blade tip clearance. Under multiple rotational speed conditions, the range of low-energy flow clusters within the passage is reduced to varying degrees. Additionally, the outlet width of the meridional passage after optimization is reduced, resulting in reduced expansion, which has a drainage effect on low-speed fluid, thereby reducing flow blockage losses and improving the stability of flow inside the centrifugal compressor.

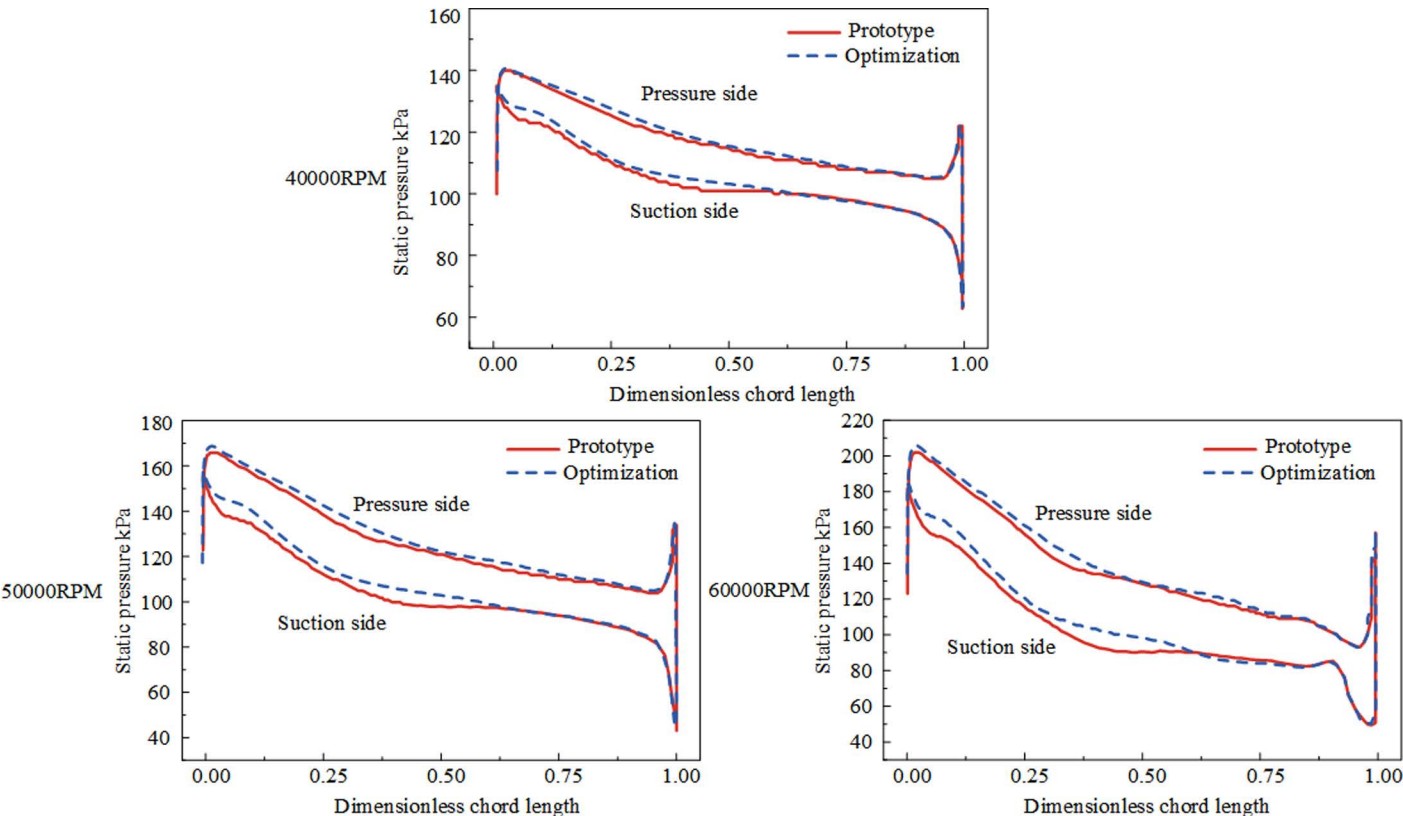

**Fig 12. The static pressure distribution of the main blade at 90% blade height under the highest efficiency condition before and after optimization.**

## 5 Conclusion

(1) Post-optimization, the meridional curvature of the impeller remains continuous from axial to radial direction, with both inlet shroud and hub profiles collectively shifting outward. The width of the inlet section undergoes minimal change, while that of the midsection expands, ensuring uniform airflow. The transition region between the shroud and hub remains consistent, enhancing flow dynamics. At the exit section of the meridional passage, both shroud and hub profiles contract inward, resulting in a reduced height of the rear section of the passage, thereby decreasing expansion and increasing gas kinetic energy to suppress internal flow separation.

(2) Post-optimization, the meridional curvature of the impeller expands the stable operating range of the centrifugal compressor by 3.93%, 2.46%, and 2.84% at 40000RPM, 50000RPM, and 60000RPM rotational speeds, respectively, compared to the prototype design. Furthermore, the efficiency improves by 0.86%, 1.00%, and 0.42% at these respective speeds. Efficiency under multi-speed conditions consistently surpasses that of the prototype design throughout the stable operating range. Additionally, the impeller's meridional passage's work area increases, and the pressure ratio marginally increases compared to the prototype design.

(3) Post-optimization, the meridional curvature of the impeller transitions smoothly from axial to radial direction. Under all three rotational speed conditions, it suppresses

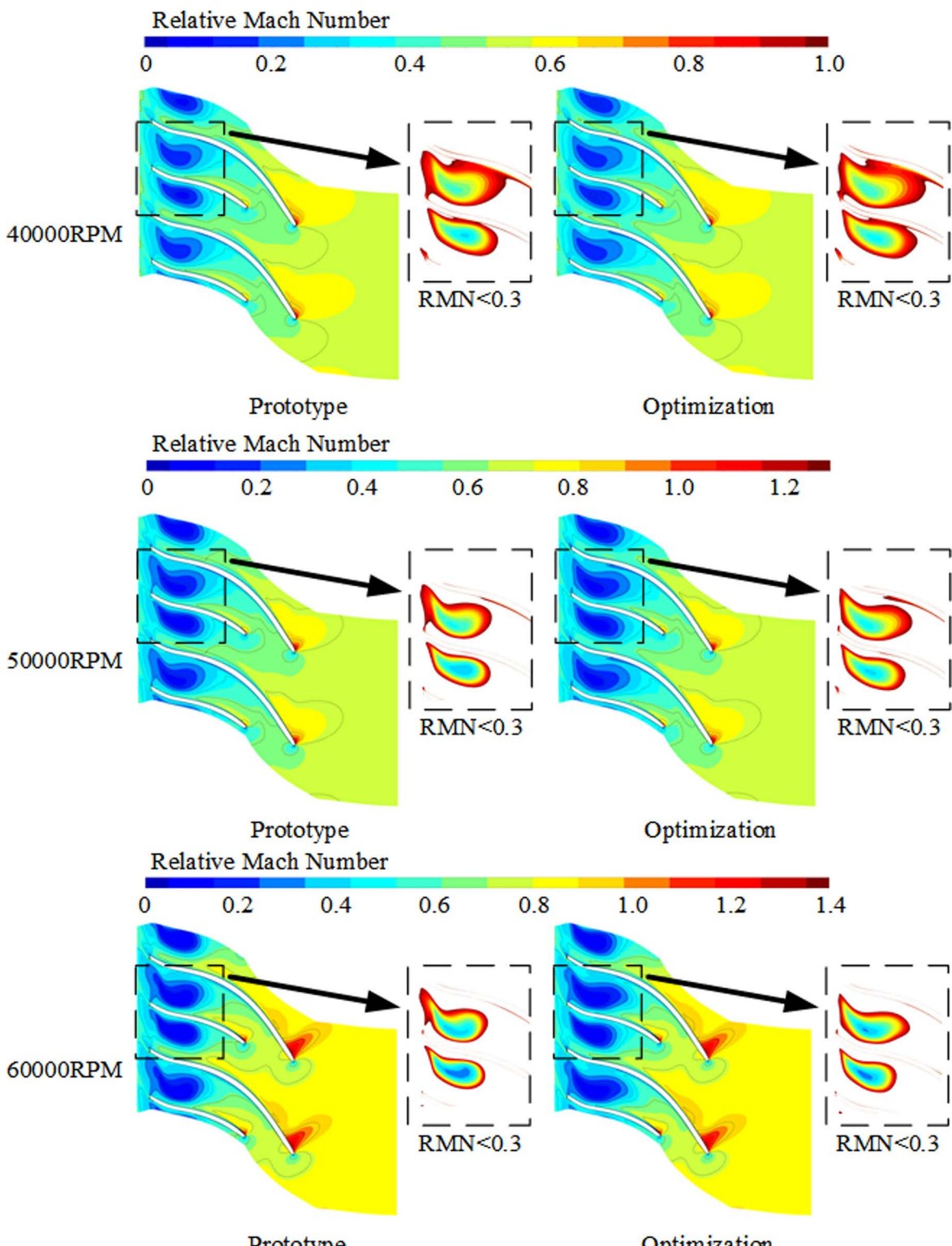

**Fig 13. The relative Mach number distribution diagram at 90% blade height under the highest efficiency condition before and after optimization.**

secondary flows near the impeller shroud and blade tip clearance leakage. Moreover, after optimization, the curvature at the passage transition remains continuous, resulting in a reduced pressure gradient along the blade height direction, facilitating smoother airflow at the transition point, and reducing entropy near the impeller shroud. The rear section of the passage sees a reduction in height, leading to a decrease in exit expansion, which aids in draining low-speed fluid, reducing airflow blockage losses, and decreasing mixing losses between low-energy and mainstream flows. The uniformity of the flow field at the impeller exit improves under multi-speed conditions compared to the prototype design.

## Author contributions

**Conceptualization:** Ning Yu, Xiaohan Yu.

**Data curation:** Xiaohan Yu, Zhi Cai, Xinle Yang.

**Investigation:** Zhi Cai, Xinle Yang.

**Methodology:** Ning Yu, Xiaohan Yu.

**Resources:** Song Li.

**Writing – original draft:** Ning Yu, Xiaohan Yu, Liyong Tian.

**Writing – review & editing:** Ning Yu, Xiaohan Yu, Liyong Tian, Song Li.

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
