## [Decision Letter · Decision Letter 0]

19 Nov 2024

PONE-D-24-45477Research on Multi-condition Optimization of Centrifugal Compressor Impeller Meridian ProfilePLOS ONE

Dear Dr. Yu,

Thank you for submitting your manuscript to PLOS ONE. After careful consideration, we feel that it has merit but does not fully meet PLOS ONE’s publication criteria as it currently stands. Therefore, we invite you to submit a revised version of the manuscript that addresses the points raised during the review process.

We look forward to receiving your revised manuscript.

Kind regards,

Hongbing Ding, Ph.D.

Academic Editor

PLOS ONE

Journal Requirements:

Please confirm at this time whether or not your submission contains all raw data required to replicate the results of your study. Authors must share the “minimal data set” for their submission. PLOS defines the minimal data set to consist of the data required to replicate all study findings reported in the article, as well as related metadata and methods (https://journals.plos.org/plosone/s/data-availability#loc-minimal-data-set-definition ).

If your submission does not contain these data, please either upload them as Supporting Information files or deposit them to a stable, public repository and provide us with the relevant URLs, DOIs, or accession numbers. For a list of recommended repositories, please see https://journals.plos.org/plosone/s/recommended-repositories .

Additional Editor Comments:

Thank you for submitting your manuscript to PLOS ONE. The reviewers recommend reconsideration of your paper following major revision. I invite you to resubmit your manuscript after addressing all reviewer comments.

Reviewers' comments:

Reviewer's Responses to Questions

**Comments to the Author**

1. Is the manuscript technically sound, and do the data support the conclusions?

Reviewer #1: Yes

Reviewer #2: Partly

2. Has the statistical analysis been performed appropriately and rigorously? 

Reviewer #1: N/A

Reviewer #2: N/A

3. Have the authors made all data underlying the findings in their manuscript fully available?

Reviewer #1: Yes

Reviewer #2: Yes

4. Is the manuscript presented in an intelligible fashion and written in standard English?

Reviewer #1: Yes

Reviewer #2: No

5. Review Comments to the Author

Reviewer #1: Based on the numerical simulation calculation and experimental verification of the performance of the prototype centrifugal compressor, the meridian profile of the prototype centrifugal compressor impeller was modified and designed, and NSGA-III optimization algorithm was used to implement multi-condition optimization for the structural parameters of the prototype impeller meridian profile. Under multi-speed conditions, the characteristic curves and internal flow field of the optimized scheme were compared with that of the prototype scheme, and the influence of the multi-condition optimization scheme on the characteristics and internal flow of the centrifugal compressor under multi-speed conditions was analyzed and discussed. The topic selection of this paper touches the frontier of the subject, and is closely combined with the engineering practice, and its research conclusion has certain theoretical reference value. Although the subject matter of this paper has some practical relevance, it also displays obvious shortcomings.Therefore, I recommend that this paper undergo minor revisions before being considered for publication.

1. There are problems such as grammatical errors, unclear ideation and improper use of the article, so it is necessary to improve the full text language.

2.The consistency of terminology must be ensured, for example, the term "meridional contour" should be used uniformly in the abstract and the whole paper.

3. When the impeller hub rim changes, the extension mode of the blade shape should be elaborated.

4. Explain why you chose the single channel calculation and why you did not use other calculation methods.

5. Within a certain speed range, compressor characteristics should be similar. If so, is it more practical to highlight the coordination of multiple operating conditions (not less than 3 points) at the design speed? On the contrary, it is necessary to emphasize the reasons for the obvious difference in characteristics when the speed changes.

Reviewer #2: The article is unclear, lacks logic, and has some sentences with incorrect expressions. From the abstract, it can be seen that the author of this article intends to focus on using the NSGA-III method, but there are few references to the NSGA-III method throughout the full text, and there is not a single reference cited in the entire third section.

6. PLOS authors have the option to publish the peer review history of their article (what does this mean? ). If published, this will include your full peer review and any attached files.

**Do you want your identity to be public for this peer review?** For information about this choice, including consent withdrawal, please see our Privacy Policy .

Reviewer #1: No

Reviewer #2: No

---

## [Author Response · Author response to Decision Letter 1]

4 Dec 2024

The opinions of the reviewers have been replied and answered one by one in the article.

---

## [Decision Letter · Decision Letter 1]

22 Jan 2025

Research on Multi-condition Optimization of Centrifugal Compressor Impeller Meridian Profile

PONE-D-24-45477R1

Dear Dr. Yu,

We’re pleased to inform you that your manuscript has been judged scientifically suitable for publication and will be formally accepted for publication once it meets all outstanding technical requirements.

Kind regards,

Hongbing Ding, Ph.D.

Academic Editor

PLOS ONE

Additional Editor Comments (optional):

The authors have done a good job in revising the manuscript. Now it can be accepted for publication in PLOS ONE.

Reviewers' comments:

Reviewer's Responses to Questions

**Comments to the Author**

1. If the authors have adequately addressed your comments raised in a previous round of review and you feel that this manuscript is now acceptable for publication, you may indicate that here to bypass the “Comments to the Author” section, enter your conflict of interest statement in the “Confidential to Editor” section, and submit your "Accept" recommendation.

Reviewer #2: All comments have been addressed

2. Is the manuscript technically sound, and do the data support the conclusions?

Reviewer #2: Yes

3. Has the statistical analysis been performed appropriately and rigorously? 

Reviewer #2: Yes

4. Have the authors made all data underlying the findings in their manuscript fully available?

Reviewer #2: Yes

5. Is the manuscript presented in an intelligible fashion and written in standard English?

Reviewer #2: Yes

6. Review Comments to the Author

Reviewer #2: While the optimization results show performance improvement, the adaptability and limitations of the optimization scheme under different working conditions can be further explored, such as the performance of the optimization scheme under extreme working conditions or special application scenarios, and how it can be further optimized to meet these challenges.Some of the graphs are optimized and designed to be more intuitive and easy to understand.

When presenting the optimization results, it is recommended to add more detailed data comparisons and analyses so that readers can have a more in-depth understanding of the effectiveness of the optimization scheme. For example, the magnitude of efficiency improvement at different rotational speeds, changes in pressure distribution, etc. can be compared.When writing the paper, it is recommended to further sort out and cite the latest research results in related fields to enhance the academic and cutting-edge nature of the paper. Attention should also be paid to checking the cited literature information to ensure accuracy and completeness.

7. PLOS authors have the option to publish the peer review history of their article (what does this mean? ). If published, this will include your full peer review and any attached files.

**Do you want your identity to be public for this peer review?** For information about this choice, including consent withdrawal, please see our Privacy Policy .

Reviewer #2: No

---

## [Editor Report · Acceptance letter]

PONE-D-24-45477R1

PLOS ONE

Dear Dr. Yu,

I'm pleased to inform you that your manuscript has been deemed suitable for publication in PLOS ONE. Congratulations! Your manuscript is now being handed over to our production team.

Kind regards,

on behalf of

Professor Hongbing Ding

Academic Editor

PLOS ONE